# Reversible Cerebral Vasoconstriction Syndrome and Raynaud’s Phenomenon: Is There a Link between the Pathogeneses of Their Underlying Complex Etiology? A Case Report and Literature Review

**DOI:** 10.3390/diagnostics13182951

**Published:** 2023-09-14

**Authors:** Fahidah Alenzi, David P D’Cruz

**Affiliations:** 1Clinical Sciences Department, College of Medicine, Princess Nourah bint Abdulrahman University, Riyadh 11564, Saudi Arabia; 2Louise Coote Lupus Unit, Guy’s Hospital, Guy’s and St. Thomas’ Hospitals, NHS Foundation Trust, London SE1 9RT, UK

**Keywords:** Raynaud’s phenomenon, reversible cerebral vasoconstriction syndrome, nail-fold capillaroscopy, autoimmune rheumatic diseases, systemic lupus erythematosus

## Abstract

Reversible cerebral vasoconstriction syndrome (RCVS) typically manifests as a sudden, severe thunderclap headache due to narrowing of the cerebral arteries. Symptoms usually resolve within three months. An imbalance in cerebral vascular tone, an abnormal endothelial function, and a decreased autoregulation of cerebral blood flow are thought to be involved in the pathogenesis of RCVS. However, the precise origin of this condition is not yet fully understood. Symptoms of Raynaud’s phenomenon (RP) include vasospasm of arterioles of the digits. The pathophysiology of RP includes interactions between the endothelium, smooth muscle, and autonomic and sensory neurons that innervate arteries to help maintain vasomotor homeostasis. RP may occur before the clinical manifestation of a rheumatic condition. RCVS is rare in patients with autoimmune rheumatic disease. We describe a 54-year-old female who had a history of Raynaud’s phenomenon affecting her fingers and toes since the age of 12 years. The patient was diagnosed with RCVS in 2012. She described RCVS precipitants, including the regular use of cannabis, cocaine, and amphetamine and tobacco smoking. In 2021, she presented with oral ulcers, intermittent swallowing difficulties, and Raynaud’s phenomenon. Clinical examination revealed early sclerodactyly, and abnormal nail-fold capillaroscopy showed multiple giant capillaries, dilated capillary loops, and areas of capillary hemorrhage with capillary drop-out. The investigation revealed positive ANA, strongly positive SRP antibodies, and Ro60 antibodies. Our case report indicates that there may be a correlation between RCVS and Raynaud’s phenomenon, and a potential connection between RCVS and autoimmune rheumatic diseases. Hence, physicians must be aware of the red flags and subtle differences in neurological abnormalities, such as headaches, in patients with autoimmune rheumatic diseases who have an inactive clinical status to improve patient care and outcomes.

## 1. Introduction

Reversible cerebral vasoconstriction syndrome (RCVS) is a rare disease characterized by severe acute thunderclap headaches associated with cerebral artery constriction (a ‘string of beads’ appearing in the cerebral arteries) that spontaneously reverses in 12 weeks and can occur at any age. RCVS may occur spontaneously; however, in many cases, a secondary factor can contribute, such as vasoactive medications or postpartum status [1,2]. An imbalance of cerebral vascular tone, an abnormal endothelial function, and a decreased autoregulation of cerebral blood flow are thought to be involved in the disease process of RCVS; however, the precise origin of this condition is not fully understood [3,4,5]. Maurice Raynaud originally characterized the Raynaud’s phenomenon (RP) in 1862 [6]. Young women and those with a positive family history of Raynaud’s phenomenon are more likely to be affected. Raynaud’s phenomenon symptoms include vasospasm of arterioles of the digits, resulting in ischemia-associated skin pallor, followed by cyanosis, and blushing from re-perfusion and are the color changes associated with Raynaud’s phenomenon, and it is possible for complicated RP with prolonged tissue ischemia to cause pain, ulceration of the digits, which may ultimately result in gangrene or digit amputation [6]. The pathophysiology of RP includes interactions between the endothelium, smooth muscle, and autonomic and sensory neurons that innervate arteries to help maintain vasomotor homeostasis. Moreover, the extent of endothelial and vascular damage is the key distinction between primary and secondary RP, with secondary RP showing more severe damage [7]. In secondary RP, injured endothelial cells become activated, promoting the production of intravascular microthrombi and procoagulant activity, as well as inducing the proliferation of smooth muscle cells and an inflammatory response. As a result, neurogenic mechanisms predominate in initial RP, while endothelial and immunological mechanisms are more significant in secondary RP. These three processes combine to cause and spread RP. As RP is a common presenting symptom of many rheumatic diseases, rheumatic disorders should be ruled out when RP first manifests [8,9,10]. RP may occur before the clinical manifestation of a rheumatic conditions [7,11]. RCVS has rarely been documented in autoimmune rheumatic diseases. In recent years, there has been growing interest in the connection between RCVS and Raynaud’s disease. Both disorders exhibit similar abnormalities in vascular reactivity, endothelial dysfunction, and sympathetic nervous system activity, which are characterized by both vasoconstriction and poor vasodilation. Endothelial dysfunction contributes to the development of both diseases [3,4,5,12,13]. Additionally, abnormalities in the release of vasoactive molecules such as endothelin-1, and dysregulation of sympathetic nervous system activity may also play a role in the etiology of Raynaud’s phenomenon and RCVS [14,15]. The possibility of cerebral artery involvement in autoimmune rheumatic diseases is not well described because of the rarity of both disorders [1,2,3,5,12,13,14,15,16,17]. RCVS was originally described using the diagnostic criteria of Calabrese et al. in 2007 [2]. They include the following: documentation of multifocal segmental cerebral artery vasoconstriction on angiographic analysis that improves in 3 months, with no evidence of subarachnoid hemorrhage (SAH) or cerebrospinal fluid pathologies, and absence of other neurological signs or symptoms with acute, severe headaches. Existing studies emphasize the heterogeneity demonstrated in patients with RCVS and how this may make it difficult to diagnose subtle forms of RCVS [17]. The best indicator of connective tissue disorders in patients with RP is an abnormal capillary nail-fold examination. Nail-fold capillaroscopy is a useful and practical procedure since it emphasizes early, rapid imaging approaches for disease diagnosis and management, and the principal indications for nail-fold capillaroscopy are usually rheumatological disorders. Over the past few decades, the diagnostic value of nail-fold capillaroscopy has improved. Furthermore, in rheumatology, nail-fold capillaroscopy is a common technique for analyzing connective tissue disorders and evaluating the primary and secondary causes. Nail-fold capillaroscopy investigation in rheumatological conditions has improved over time to the point where it is now included in the diagnostic criteria for Raynaud’s phenomenon [4]. Nail-fold capillaroscopy has emerged as a useful diagnostic technique because it is a noninvasive method that assesses and detects microvascular abnormalities in nail-fold beds, morphology, and capillary distribution, such as capillary dilatation, capillary dropout, and tortuosity, which contribute to the existence of autoimmune rheumatic diseases such as systemic sclerosis, mixed connective disease, dermatomyositis, and systemic lupus erythematosus [18]. One of the diagnostic characteristics of primary Raynaud’s syndrome is the absence of an abnormal capillaroscopic pattern. Systemic sclerosis, systemic lupus erythematosus, rheumatoid arthritis, Sjogren’s syndrome, and dermatomyositis are among the rheumatologic disorders associated with secondary Raynaud’s phenomenon [4]. Furthermore, in the setting of RCVS and Raynaud’s syndrome, nail-fold capillaroscopy may reveal distinctive patterns, such as large capillaries or avascular zones, which can assist in the diagnosis and monitoring of microvascular involvement [18]. Therefore, these patterns may be helpful. In addition, longitudinal studies that use nail-fold capillaroscopy may assist in the identification of possible predictors of the course of the disease and the patient’s response to therapy. Few case reports [1,5,14,15,16], as shown in Table 1, and retrospective investigations have reported patients with SLE who developed RCVS or signs similar to RCVS [19]. It is possible that both diseases may be caused, in part, by similar pathological mechanisms such as dysfunction of the endothelium, immunological disorders, and vasospasm. Given its potential impact on diagnostic and therapeutic procedures, it is crucial for physicians to be aware of the possibility of a correlation between RCVS and SLE. We report a case of a patient who was initially diagnosed with reversible cerebral vasoconstriction syndrome (RCVS), but later exhibited clinical, laboratory, and capillaroscopic findings consistent with an underlying autoimmune rheumatic disorder.

## 2. Case Presentation

A 54-year-old female patient presented with a history of autism, Asperger’s syndrome, attention deficit hyperactivity disorder, complex post-traumatic stress disorder, and Raynaud’s phenomenon affecting her fingers and toes since she was 12 years old. She developed psoriasis at 10 years of age. She had undergone silicone breast implants 20 years ago. In 2012, she presented with severe thunderclap headaches and was diagnosed by neurologists with reversible cerebral vasoconstriction syndrome.

In 2021, she developed fatigue, oral ulcers, Raynaud’s phenomenon, difficulty swallowing, intermittent blurred vision, poor handwriting, nausea, difficulty walking, and depression. There was no history of digital, oral, or genital ulcers, miscarriages, or thrombosis. She reported regular cannabis and tobacco use and alcohol consumption. She had been a cocaine and amphetamine user in the past. She had a family history of Raynaud’s phenomenon. On examination, she had normal blood pressure, early sclerodactyly, faint livedo reticularis on her lower limbs, minor lymphadenopathy in the neck and axillae, an ejection systolic murmur, and an early diastolic murmur in accordance with the aortic valve lesion on the echocardiogram. Chest, abdomen, neurology, and musculoskeletal examinations were normal. The investigation showed positive antinuclear antibody (ANA) of 1:80, positive anti-signal recognition particle (SRP) antibodies, positive Ro/SS-A (Ro60) antibodies, negative anti-double stranded DNA (anti-dsDNA), antineutrophil cytoplasmic antibodies (ANCA), rheumatoid factor, and antiphospholipid antibodies, normal complement, normal CK level, troponin, brain natriuretic peptide (BNP), and inflammatory markers, and negative urinalysis. The urine screening was positive for cannabis and negative for cocaine.

Nail-fold capillaroscopy was abnormal with multiple giant capillaries and dilated capillary loops, and many areas of capillary hemorrhage with some capillary drop-out were evident (Figure 1).

## 3. Discussion

There may be an association between RCVS and Raynaud’s phenomenon in patients with systemic autoimmune disorders [1]. Secondary precipitants of RCVS have been reported in at least half of patients, including exposure to recreational drugs or vasoactive medications such as cannabis, cocaine, ecstasy, amphetamines, lysergic acid diethylamide (LSD), nicotine patches, ergots, alcohol drinking, decongestants, electronic cigarettes, and beta-blockers [1,2,5,13,16]. Other conditions include the postpartum state, trauma, surgery, eclampsia, cervical artery dissection, subdural hematoma, blood product exposure, catecholamine-secreting tumors, cerebral vasculitis, autonomic dysfunction, and microangiopathic hemolytic anemia [1,2,13,14,15,16,17,18]. RCVS can be precipitated by genetic and environmental factors that cause abnormal oxidative stress, endothelial dysfunction, and sympathetic overactivity. Studies suggest that Raynaud’s phenomenon may affect almost all blood vessels, including smooth muscles, and for this reason, it could be considered a systemic process. In our case, although it was not certain if RCVS was linked to autoimmune rheumatic diseases, her clinical symptoms of Raynaud’s phenomenon, oral ulcers, intermittent swallowing difficulties, possible sclerodactyly, positive autoantibodies, and abnormal capillaroscopy suggested an underlying autoimmune rheumatic disorder. However, a correlation between RCVS and SLE has rarely been reported. A similar case was evident in a study by Sayegh et al., 2010 [1], who reported a patient with SLE who was initially misdiagnosed with lupus-associated cerebral vasculitis, but on subsequent investigation, was diagnosed with RCVS. The study noted that RCVS might be difficult to distinguish from cerebral vasculitis or other similar autoimmune CNS disorders on initial evaluation. Both conditions may develop acute severe headaches, localize neurological impairments, and cause distinctive abnormalities in brain images [20]. Moreover, vasculitis is associated with persistent inflammation with insidious-onset headache, which progresses gradually [21]. In contrast, RCVS is associated with acute ‘thunderclap headaches,’ which are self-limiting and a typical characteristic, as demonstrated in our case [1]. Similar studies have emerged that support the probable connection between RCVS and autoimmune rheumatological disorders such as SLE [5,12,13,14,15,17,18,20,21,22]. This suggests that the two conditions may have similar underlying pathogenic processes. It is possible that abnormal vascular reactivity, malfunction of the endothelium, and dysregulation of the immune system are all factors that contribute to the development of these disorders. Due to the fact that our patient had RVCS and Raynaud’s phenomenon, the issue arises as whether or not there is a causal connection between the two conditions. Although the surgical history of breast implants in our case was not directly linked with RCVS in SLE, studies have also reported the use of immunosuppressant drugs, especially tocilizumab, prednisolone, and cyclophosphamide, and blood transfusion as a rare yet potential cause of RCVS when used as a treatment for connective tissue diseases [19,20,21,22]. Studies have also revealed the presence of RCVS in pediatric patients with SLE following immunosuppressive treatment for lupus nephritis, thereby mandating its early recognition, which may help adjust further treatment [23,24]. Calcineurin inhibitors have been identified as triggers for RCVS in the context of systemic lupus erythematosus (SLE), as they can cause damage to the endothelium of cerebral arteries and disrupt vascular tone control [19,20,21,22]. Moreover, existing data suggest that RCVS shares many common features with inflammatory CNS disorders that often require careful interpretation [22]. There may be common pathways between RCVS and autoimmune rheumatic illnesses, and to achieve an accurate diagnosis and determine the course of therapy, with evidence from our case, we plan to consider MRI and EMG of the thigh muscles and esophageal manometry to further investigate her swallowing difficulties despite normal CK levels and the absence of any muscular symptoms. The use of nail-fold capillaroscopy is crucial for prognostication and for avoiding serious consequences. In our case, we used the parameters evaluated during the capillaroscopic examination, and the pathologic findings included the morphology of the capillaries, such as enlarged capillaries, capillary tortuosity, enlarged capillaries, giant capillaries, the presence of pericapillary edema, microhemorrhage, avascular areas, and neo-angiogenesis. Systemic lupus erythematosus nail-fold capillaroscopy alterations have been reported to correspond to disease activity and systemic symptoms [25]. Furthermore, there is no available information regarding the use of nail-fold capillaroscopy for disorders other than connective tissue diseases in determining disease activity and correlating with other disease-related factors [26]. The patient was given a treatment with ramipril 2.5 mg daily, verapamil 240 mg daily, mirtazapine 15 mg daily. However, studies [5] have reported that further treatment with oral nimodipine may help in the recovery of these patients. Although treatments with hydroxychloroquine and immunomodulatory therapies are indicated to improve clinical features, evidence is contradictory for immunosuppressants. It is also important to avoid common precipitants of RCVS, such as cocaine, amphetamine, and cannabis, which may induce vasoconstriction [13]. The diagnosis and therapy of RCVS within the setting of autoimmune rheumatic illnesses may benefit from an understanding of the common immunological pathways, and the identification of particular autoantibodies or biomarkers, as a careful deliberation, is required while treating RCVS in individuals with autoimmune rheumatic disorders.

## 4. Conclusions

We report a patient who was initially diagnosed with reversible cerebral vasoconstriction syndrome but ultimately exhibited clinical, serological, and capillaroscopic evidence for an underlying autoimmune rheumatic disorder. Our case report suggests that there may be an association between RCVS and Raynaud’s phenomenon, and a possible link between RCVS and autoimmune rheumatic diseases. The use of nail-fold capillaroscopy is crucial for prognostication and for avoiding serious consequences. RCVS diagnosis in patients with autoimmune rheumatic disease is a key point that prompts rapid treatment and prognosis. Hence, physicians must be aware of the red flags and subtle differences in neurological abnormalities, such as headaches, in patients with autoimmune rheumatic diseases who have an inactive clinical status to improve patient care and outcomes.

## Figures and Tables

**Figure 1 diagnostics-13-02951-f001:**
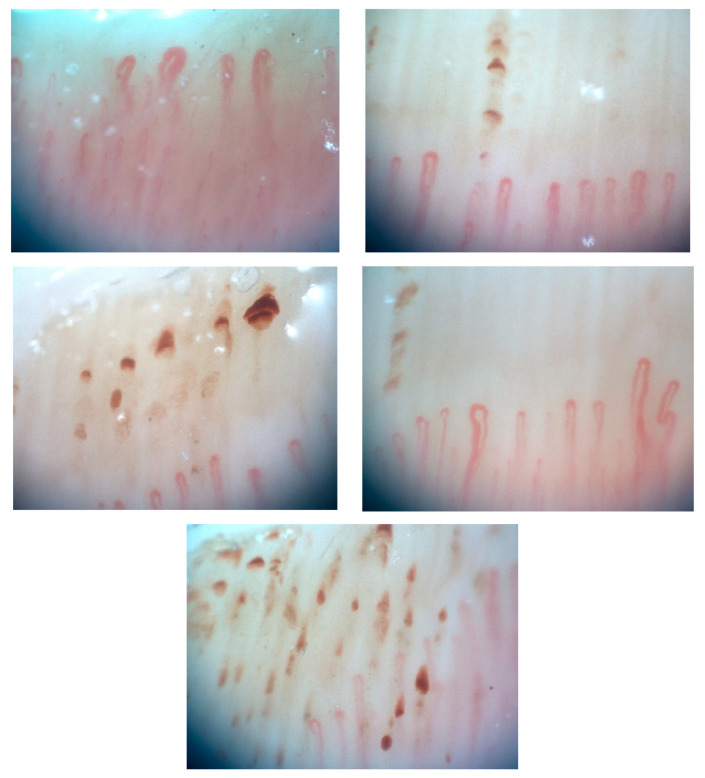
Nail-fold capillaroscopy shows multiple giant capillaries, dilated capillary loops, and areas of capillary hemorrhage with capillary drop-out. Patient consent was obtained.

**Table 1 diagnostics-13-02951-t001:** Summary of reported cases of autoimmune rheumatic disease and RCVS.

Author	Patient Gender	Ethnicity	Age at Presentation	Rheumatic Disease Association	Journal	Year of Publication	Ref.
Liu et al.	Female	African	44 years	Systemic sclerosis	Cureus	2022	[14]
Etemadifar et al.	Female	Not stated	35 years	Systemic sclerosis	Case Reports in Immunology	2022	[15]
Chung et al.	Female	Not stated	35 years	systemic lupus erythematosus	Lupus	2019	[5]
Ashraf et al.	Female	Not stated	42 years	systemic lupus erythematosus	Neurology India	2012	[16]
Sayegh et al.	Female	Not stated	40 years	systemic lupus erythematosus	Rheumatology	2010	[1]

## Data Availability

Further inquiries can be directed to the corresponding author.

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
