# Peer review of "Reversible Cerebral Vasoconstriction Syndrome and Raynaud’s Phenomenon: Is There a Link between the Pathogeneses of Their Underlying Complex Etiology? A Case Report and Literature Review"

_diagnostics, 2023, doi:10.3390/diagnostics13182951_

Round 1
Reviewer 1 Report
The case report is interesting
I have some comments:
1- Abstract. Our case report suggests that there may be an association between RCVS and Raynaud’s phenomenon, and a possible link between RCVS and autoimmune rheumatic diseases. Please improve this sentence and underline the novelty of this interesting case report.
2- Introduction. Young women and families of patients with Raynaud’s phenomenon are more 48 likely to experience Raynaud's phenomenon. Please re-write this sentence.
3- Introduction. As RP is a common presenting symptom of many rheu- 63 matic diseases, rheumatic disorders should be ruled out when RP first manifests. Likewise, some additional references could complement the text:
a- Diagnostics 2023, 13, 2204. Methods of Assessing Nailfold Capillaroscopy Compared to Video Capillaroscopy in Patients with Systemic Sclerosis—A Critical Review of the Literature. https://doi.org/10.3390/diagnostics13132204
b- Front Pharmacol. 2019;10:360. Innovations in the Assessment of Primary and Secondary Raynaud's Phenomenon. doi:10.3389/fphar.2019.00360
c- J Rheumatol. 2016;43(3):599-606. Quantitative Alterations of Capillary Diameter Have a Predictive Value for Development of the Capillaroscopic Systemic Sclerosis Pattern. doi:10.3899/jrheum.150900.
4) We describe a patient who was initially diagnosed with reversible cere- 111 bral vasoconstriction syndrome but ultimately developed clinical, serological, and capil- 112 laroscopic evidence for an underlying autoimmune rheumatic disorder. Improve this sentence and underline the novelty of this case report.
5) Figure 1. Nail Fold capillaroscopy shows multiple giant capillaries, dilated capillary loops and areas 140 of capillary haemorrhage with capillary drop-out. Patient consent obtained’. Improve the quality of figures. Please, ameliorate the legend of figures. Add the magnification of the figures.
6) 3. Discussion 143 There may be an association between RCVS and Raynaud’s phenomenon in patients 144 with systemic autoimmune disorders [1]. summarized here the peculiarities of the clinical case.
7) 4. Conclusions 214 We report a patient who was initially diagnosed with reversible cerebral vasocon- 215 striction syndrome but ultimately developed clinical, serological, and capillaroscopic ev- 216 idence for an underlying autoimmune rheumatic disorder. Our case report suggests that 217 there may be an association between RCVS and Raynaud’s phenomenon, and a possible 218 link between RCVS and autoimmune rheumatic diseases. The use of nail-fold capillaros- 219 copy is crucial for prognostication and for avoiding serious consequences. RCVS diagno- 220 sis in patients with autoimmune rheumatic disease is a key point that prompts rapid treat- 221 ment and prognosis. Hence, physicians must be aware of the red flag signs and subtle 222 differences in neurological abnormalities, such as headaches, in patients with autoim- 223 mune rheumatic diseases who have an inactive clinical status to improve patient care and 224 outcomes. Underline here the peculiarities of the clinical case and highlight how this case can be useful in the clinical practice of other doctors.
Author Response
Author’s Response: Our esteemed Reviewer’s suggestion was carefully discussed.
We appreciate the thorough evaluation of the manuscript by the Reviewers and are grateful to them for improving the paper.
Reviewer 1 Comment
I have some comments:
- Our case report suggests that there may be an association between RCVS and Raynaud’s phenomenon, and a possible link between RCVS and autoimmune rheumatic diseases. Please improve this sentence and underline the novelty of this interesting case report.
Author’s Response: We improve the understanding of this sentence by writing:
Our case report indicates that there may be a correlation between RCVS and Raynaud's phenomenon, and a potential connection between RCVS and autoimmune rheumatic diseases.
Author’s Response: The novelty of this interesting case report is written at the end of the last question.
- Young women and families of patients with Raynaud’s phenomenon are more 48 likely to experience Raynaud's phenomenon. Please re-write this sentence.
Author’s Response: We improve the understanding of this sentence by writing:
Young women and those with a positive family history of Raynaud's are more likely to be affected.
3- Introduction. As RP is a common presenting symptom of many rheu- 63 matic diseases, rheumatic disorders should be ruled out when RP first manifests. Likewise, some additional references could complement the text:
- Diagnostics2023, 13, 2204. Methods of Assessing Nailfold Capillaroscopy Compared to Video Capillaroscopy in Patients with Systemic Sclerosis—A Critical Review of the Literature. https://doi.org/10.3390/diagnostics13132204
- Front Pharmacol. 2019;10:360. Innovations in the Assessment of Primary and Secondary Raynaud's Phenomenon. doi:10.3389/fphar.2019.00360
- J Rheumatol. 2016;43(3):599-606. Quantitative Alterations of Capillary Diameter Have a Predictive Value for Development of the Capillaroscopic Systemic Sclerosis Pattern. doi:10.3899/jrheum.150900.
Author’s Response: We acknowledge this recommendation, these suggested references are included and cited.
4) We describe a patient who was initially diagnosed with reversible cere- 111 bral vasoconstriction syndrome but ultimately developed clinical, serological, and capil- 112 laroscopic evidence for an underlying autoimmune rheumatic disorder. Improve this sentence
Author’s Response: We report a case of a patient who was initially diagnosed with reversible cerebral vasoconstriction syndrome (RCVS), but later developed clinical, laboratory, and capillaroscopic findings consistent with an underlying autoimmune rheumatic disorder.
5) Figure 1. Nail Fold capillaroscopy shows multiple giant capillaries, dilated capillary loops and areas 140 of capillary haemorrhage with capillary drop-out. Patient consent obtained’. Improve the quality of figures. Please, ameliorate the legend of figures. Add the magnification of the figures.
Author’s Response: We acknowledge your comment, but these figures were taken from the capillaroscopy machine in the clinic after examining our reported patient, and there was no magnification for it.
6) 3. Discussion 143 There may be an association between RCVS and Raynaud’s phenomenon in patients 144 with systemic autoimmune disorders [1]. summarized here the peculiarities of the clinical case.
7) 4. Conclusions 214 We report a patient who was initially diagnosed with reversible cerebral vasocon- 215 striction syndrome but ultimately developed clinical, serological, and capillaroscopic ev- 216 idence for an underlying autoimmune rheumatic disorder. Our case report suggests that 217 there may be an association between RCVS and Raynaud’s phenomenon, and a possible 218 link between RCVS and autoimmune rheumatic diseases. The use of nail-fold capillaros- 219 copy is crucial for prognostication and for avoiding serious consequences. RCVS diagno- 220 sis in patients with autoimmune rheumatic disease is a key point that prompts rapid treat- 221 ment and prognosis. Hence, physicians must be aware of the red flag signs and subtle 222 differences in neurological abnormalities, such as headaches, in patients with autoim- 223 mune rheumatic diseases who have an inactive clinical status to improve patient care and 224 outcomes. Underline here the peculiarities of the clinical case and highlight how this case can be useful in the clinical practice of other doctors.
Author’s Response: We acknowledge your comment In Q6 and Q7
Author answer for Q 1, Q6 and Q7:
The peculiarities of this interesting case report and its novelty as follow and it is mentioned throughout the manuscript:
1-Given its potential impact on diagnostic and therapeutic procedures, it is crucial for physicians to be aware of the possibility of a correlation between RCVS and SLE
2-The diagnosis and therapy of RCVS within the setting of autoimmune rheumatic illnesses may benefit from an understanding of the common immunological pathways, and the identification of particular autoantibodies.
3-There may be common pathways between RCVS and autoimmune rheumatic illnesses and to achieve an accurate diagnosis and determine the course of therapy, physician should investigate the patient extensively as early recognition, will help in adjusting further treatment and prognosis.
4-The possible pathophysiological factors that contribute to the development of both RCVS and Raynaud’s phenomenon are abnormal vascular reactivity, malfunction of the endothelium, and dysregulation of the immune system which suggests that the two conditions may have similar underlying pathogenic processes.
5-The importance of utilization of nail-fold capillaroscopy and the use of nail-fold capillaroscopy is crucial for prognostication and for avoiding serious consequences of disease.
6-Physicians must be aware of the red flag signs and subtle differences in neurological abnormalities, such as headaches, in patients with autoimmune rheumatic diseases who have an inactive clinical status to improve patient care and outcomes.
We hope our paper has now been changed following the recommendations suggested .
We look forward to a positive answer from you.
With our best regards
Sincerely Yours
Reviewer 2 Report
Correct the reference list according to the journal required format.
Author Response
Author’s Response: Our esteemed Reviewer’s suggestion was carefully discussed.
We appreciate the thorough evaluation of the manuscript by the Reviewers and are grateful to them for improving the paper.
Reviewer 2 comment
Correct the reference list according to the journal required format.
Author’s Response: We acknowledge your comment, we checked the references according to the journal format.
We hope our paper has now been changed following the recommendations suggested.
We look forward to a positive answer from you.
With our best regards
Sincerely Yours
Reviewer 3 Report
Very Respected Authors,
Abstract is well written as well as the Introduction of the manuscript. The objective is clear, methodology is well described. The description of the case is described in detail. The conclusion is in agreement with the objective and with the findings. The references are new and appropriate.
Author Response
Author’s Response:
We appreciate the thorough evaluation of the manuscript by the Reviewer 3 and are grateful to them for improving the paper.
Thank you for the positive answer from you.
With our best regards
Sincerely Yours
Reviewer 4 Report
The authors present a case report and literature review that aims to investigate the association between cerebral vasoconstriction syndrome (RCVS) and Raynaud’s phenomenon, and a possible link between RCVS and autoimmune rheumatic diseases
Comments:
If nailfold capillaroscopy has emerged as a useful diagnostic technique because it is a noninvasive method, are there comparable normal healthy images available that can be used to facilitate the identification of differences?
Author Response
Author’s Response: Our esteemed Reviewer’s suggestion was carefully discussed.
We appreciate the thorough evaluation of the manuscript by the Reviewers and are grateful to them for improving the paper.
Reviewer 3 comment
If nailfold capillaroscopy has emerged as a useful diagnostic technique because it is a noninvasive method, are there comparable normal healthy images available that can be used to facilitate the identification of differences?
Author’s Response: We acknowledge your comment.
Nailfold capillaroscopy has emerged as a useful diagnostic technique because it is simple and a noninvasive method where it describes the morphological characteristics of the capillary, enlargement and microhemorrhage.
As this is a case report, we do not compare it with normal healthy image and for our patient we do not have a normal nailfold capillaroscopy finding for her.
We hope our paper has now been changed following the recommendations suggested.
We look forward to a positive answer from you.
With our best regards
Sincerely Yours
Round 2
Reviewer 1 Report
The manuscript has been improved as requested. I have non further comments.
The manuscript is quite well written, minor changes of English language are required